# A Pilot Study of Short-Course Oral Vitamin A and Aerosolised Diffuser Olfactory Training for the Treatment of Smell Loss in Long COVID

**DOI:** 10.3390/brainsci13071014

**Published:** 2023-06-30

**Authors:** Tom Wai-Hin Chung, Hui Zhang, Fergus Kai-Chuen Wong, Siddharth Sridhar, Tatia Mei-Chun Lee, Gilberto Ka-Kit Leung, Koon-Ho Chan, Kui-Kai Lau, Anthony Raymond Tam, Deborah Tip-Yin Ho, Vincent Chi-Chung Cheng, Kwok-Yung Yuen, Ivan Fan-Ngai Hung, Henry Ka-Fung Mak

**Affiliations:** 1Department of Microbiology, Li Ka Shing Faculty of Medicine, The University of Hong Kong, Hong Kong, China; 2Department of Rehabilitation Sciences, The Hong Kong Polytechnic University, Hong Kong, China; 3Research Institute for Intelligent Wearable Systems, The Hong Kong Polytechnic University, Hong Kong, China; 4Department of Ear, Nose and Throat, Pamela Youde Nethersole Eastern Hospital, Hong Kong, China; 5State Key Laboratory of Emerging Infectious Diseases, The University of Hong Kong, Hong Kong, China; 6Carol Yu Centre for Infection, The University of Hong Kong, Hong Kong, China; 7Department of Psychology, The University of Hong Kong, Hong Kong, China; 8State Key Laboratory of Brain and Cognitive Sciences, The University of Hong Kong, Hong Kong, China; 9Department of Surgery, Li Ka Shing Faculty of Medicine, The University of Hong Kong, Hong Kong, China; 10Department of Medicine, Li Ka Shing Faculty of Medicine, The University of Hong Kong, Hong Kong, China; 11The Collaborative Innovation Center for Diagnosis and Treatment of Infectious Diseases, The University of Hong Kong, Hong Kong, China; 12Department of Diagnostic Radiology, Li Ka Shing Faculty of Medicine, The University of Hong Kong, Hong Kong, China; 13Alzheimer’s Disease Research Network, The University of Hong Kong, Hong Kong, China

**Keywords:** long COVID, smell loss, olfactory dysfunction, anosmia, vitamin A, aerosolised, olfactory training, resting-state fMRI, functional brain network

## Abstract

**Background:** Olfactory dysfunction (OD) is a common neurosensory manifestation in long COVID. An effective and safe treatment against COVID-19-related OD is needed. **Methods:** This pilot trial recruited long COVID patients with persistent OD. Participants were randomly assigned to receive short-course (14 days) oral vitamin A (VitA; 25,000 IU per day) and aerosolised diffuser olfactory training (OT) thrice daily (combination), OT alone (standard care), or observation (control) for 4 weeks. The primary outcome was differences in olfactory function by butanol threshold tests (BTT) between baseline and end-of-treatment. Secondary outcomes included smell identification tests (SIT), structural MRI brain, and serial seed-based functional connectivity (FC) analyses in the olfactory cortical network by resting-state functional MRI (rs–fMRI). **Results:** A total of 24 participants were randomly assigned to receive either combination treatment (*n* = 10), standard care (*n* = 9), or control (*n* = 5). Median OD duration was 157 days (IQR 127–175). Mean baseline BTT score was 2.3 (SD 1.1). At end-of-treatment, mean BTT scores were significantly higher for the combination group than control (*p* < 0.001, MD = 4.4, 95% CI 1.7 to 7.2) and standard care (*p* = 0.009) groups. Interval SIT scores increased significantly (*p* = 0.009) in the combination group. rs–fMRI showed significantly higher FC in the combination group when compared to other groups. At end-of-treatment, positive correlations were found in the increased FC at left inferior frontal gyrus and clinically significant improvements in measured BTT (r = 0.858, *p* < 0.001) and SIT (*r* = 0.548, *p* = 0.042) scores for the combination group. **Conclusions:** Short-course oral VitA and aerosolised diffuser OT was effective as a combination treatment for persistent OD in long COVID.

## 1. Introduction

Coronavirus disease 2019 (COVID-19) is caused by severe acute respiratory syndrome coronavirus 2 (SARS-CoV-2) [1,2]. Protracted symptoms and long-term sequelae in post-acute COVID-19 patients have since been recognised as a distinct entity called post-COVID-19 syndrome (long COVID) [3].

Olfactory dysfunction (OD) is a common neurosensory manifestation in long COVID patients [4,5]. The prevalence of persistent OD in long COVID patients is estimated to range from 11% to 57.6% [6,7]. Qualitative smell disturbances, such as phantosmia and parosmia, may also persist [8]. In the absence of effective treatment for COVID-19-related OD, outcomes of patients suffering from persistent and severe OD remain uncertain, overall worrisome of permanent neurosensory disabilities and impaired quality of life.

To augment the olfactory neurorehabilitation process, vitamin A (VitA) metabolites have been shown to regulate stem cell fate determination at the olfactory neuroepithelium (ONE) [9,10]. Multiple human studies have investigated the therapeutic role of VitA in treating OD (Appendix A) [11,12,13,14,15]. However, only one randomised control trial (RCT) was reported, where no benefits were found after three months of daily systemic treatment [14]. Crucially, olfactory training (OT), the first-line treatment for OD, was not included as a component of smell treatment in this RCT, which may have deprived the study subjects of necessary olfactory stimulations during smell recovery [16,17].

In targeting COVID-19-related OD, several interventional trials have been performed, but the trial methodologies and outcome assessments were variable (Appendix A) [18,19,20,21,22]. Overall, the current level of evidence is insufficient to provide clear clinical guidance for the management of persistent OD in long COVID patients. 

This trial was designed to evaluate the therapeutic efficacy of short-course oral VitA in combination with OT, delivered via novel aerosolisation diffusers, as an innovative treatment strategy in the management of persistent OD in long COVID patients. 

The primary objective was to determine the clinical improvements of olfactory function in combination treatment when compared to OT alone (standard care) or control (clinical observation). Secondary outcomes aimed to explore the mechanisms underpinning the clinically measurable improvements in smell function with serial multimodal magnetic resonance imaging (MRI) brain scans, structural MRI and MR spectroscopy of olfactory organs, targeted MRI evaluations of the olfactory bulbs (OB)/tracts, and serial resting–state functional MRI (rs–fMRI) brain scan assessments of the neural activity changes in the functional olfactory networks. 

## 2. Materials and Methods

### 2.1. Study Design

This was a pilot, randomised, controlled, open-label trial in long COVID patients who presented with persistent OD. The study was conducted in two tertiary medical centres in Hong Kong. The aim of the trial was to assess the therapeutic efficacy of short–course oral VitA in combination with aerosolised diffuser OT (thrice daily) for 4 weeks (combination), compared to OT alone (standard care) and clinical observation (control) for 4 weeks, in achieving smell recovery in long COVID patients. The trial protocol was approved by the institutional review boards (IRB) at both trial sites (HKECREC-2020-081, and HKU/HA HKW IRB–UW 20-454). Written informed consent was obtained from all trial participants.

### 2.2. Participants

Adult participants (aged ≥18 years) who had a history of reverse transcription-polymerase chain reaction (RT-PCR) confirmed SARS-CoV-2 infection and presented with persistent (≥3 months) OD were eligible for recruitment. Exclusion criteria included contraindications to VitA or OT, pregnancy, and pre-existing rhinological conditions. Full eligibility criteria are provided in the Appendix A (p. 24).

### 2.3. Randomisation and Blinding

Eligible participants were randomly assigned (1:1:0.5) to receive either oral VitA in combination with OT (combination group), OT alone (standard care group), or clinical observation (control group). Random assignment was unstratified and performed by independent research staff. Serial numbers linked to a computer-generated randomisation list were assigned to individual trial participants for random group allocation. Trial participants and the attending otolaryngologist (F.K.-C.W.) were not blinded to the allocated treatment. Neuroradiologists (H.Z. and H.K.-F.M.) were blinded to the group assignment and clinical outcome data during the trial.

## 3. Procedures

### 3.1. Therapeutic Interventions

Patients assigned to the combination group received short-course (14 days) oral VitA 25,000 IU (retinyl palmitate; Carlson Laboratories, Arlington Heights, IL, USA) soft gels daily in combination with OT, delivered via aerosolisation diffuser units thrice daily for 4 weeks, as previously described [11]. In brief, OT consisted of sequential exposures to four aromatic essential oils, delivered via individual aerosolisation diffuser units (SOVOS Aromatherapy, Hong Kong, China; Appendix A): lemon; eucalyptus; geranium; and cedarwood [11]. OT was conducted three times per day for 4 weeks. During OT, study participants received 20 s of odorant exposures from each category of aerosolised essential oils, therefore achieving aromatic stimulation for 80 s per treatment session. 

Patients assigned to the standard care group received identical OT treatments as described in the combination group, but without exposure to VitA. Patients assigned to the control group did not receive any intervention during the study period. 

### 3.2. Baseline Assessments

Baseline assessments included demographic characteristics; severity, characteristics, and duration of OD; risk factors; medical histories; and laboratory data. 

### 3.3. Otolaryngological and Functional Olfactory Assessments

Trial participants were screened and assessed by one specialist otolaryngologist (F.K.-C.W.) between 14 August 2020 and 11 June 2021. At enrolment, all participants received complete ear, nose, and throat assessment; nasoendoscopy examination; and quantitative olfactory function tests: butanol threshold test (BTT; prepared in-house; Appendix A) and smell identification test (SIT, University of Pennsylvania Smell Identification Test (UPSIT^®®^); Sensonics International, Haddon Heights, NJ, USA [4,23,24]). In SIT, olfactory performances of trial participants were semi-quantitatively categorised into: normosmia; microsmia (mild, moderate, severe); and anosmia (smell loss).

### 3.4. Magnetic Resonance Imaging Assessments

After randomisation, all eligible trial participants underwent MRI brain scans at baseline, week 2 (interim) and week 4 (end-of-treatment), using a 3 Tesla MR scanner (SIGNA™ Premier; GE Healthcare, Chicago, IL, USA) with a standard 48-channel head coil. 

#### 3.4.1. Structural Sequence Acquisition

Structural MR images were acquired using fast and high-resolution three-dimensional (3D) sequence (BRAVO 3D sagittal, TR = 900 ms, TI = 900 ms, Flip angle = 8°, voxel size = 1 × 1 × 1 mm^3^, FOV = 256 × 256 mm^2^). 

#### 3.4.2. Volumetric Measurements of the Olfactory Bulbs and Tracts

During MRI, sagittal 3D T2 FLAIR (TR = 6300 ms, TE = 205 ms, TI = 1812 ms, FOV = 24 cm) 1.2 mm interleaved scans were acquired. After reformatting, the coronal planes were obtained with 1.5 mm spacing. 3D volumetric measurements of bilateral OB/tracts were performed via GE workstation. The OB was anatomically defined at the anterior cribriform plate, where the olfactory tract extended posteriorly to enter the brain below the rostrum of the corpus callosum [25].

### 3.5. Magnetic Resonance Spectroscopy

MR spectroscopy was performed using the single voxel point resolved spectroscopy (PRESS; TE = 144 ms, TR = 1500 ms, voxel size = 2 × 2 × 2 cm^3^) at the gyrus rectus (GR) and superior frontal cortex (SFC). The detectable *N*–acetylaspartate/creatine (NAA/Cr) ratio by MR spectroscopy served to represent the functional neuronal integrity of the central nervous system [26,27].

### 3.6. Resting-State Functional MRI Data Acquisition and Seed–Based Analyses within the Olfactory Cortical Network

rs–fMRI brain images were collected using a gradient-echo echo-planar sequence (TE = 30 ms, TR = 2000 ms, flip angle = 80°, voxel size = 3 × 3 × 4 mm^3^) sensitive to blood-oxygen-level-dependent (BOLD) contrast. 

Pre-processing of rs–fMRI data were performed using DPABI toolbox (http://rfmri.org/dpabi; accessed on 1 August 2021) based on the SPM12 software (https://www.fil.ion.ucl.ac.uk/spm/software/spm12/; accessed on 1 August 2021). Head motion corrections were performed to adjust images to the same position. Images were excluded from further analyses if head movements were greater than 3 mm in any dimensional planes or over 3° deviation. Nuisance signals, including Friston-24 head motion parameters, mean white matter, and cerebrospinal fluid time series within brain masks, were regressed out from the time courses in each voxel. Subsequently, images were spatially normalised to the standard Montreal Neurological Institute (MNI) space and resampled to 3 × 3 × 3 mm^3^ using transformation parameters, which were estimated through DARTEL segmentation [28]. After normalisation, data were band-pass filtered (0.01 < f < 0.1 Hz) to reduce high-frequency respiratory and low-frequency cardiac noise drifts.

The hypothesis-driven region of interest (ROI) approach was applied. We defined the seed regions for functional connectivity (FC) analyses with a sphere of 10 mm radius (Appendix A). The centres of the seed regions were located at bilateral caudate nuclei [CN; MNI coordinates: left (−11, 11, 9) and right (15, 12, 9)] [29]. Subsequently, we calculated correlations between the ROI series and the whole brain for each trial participant in a voxel-wise manner. To normalise the distribution of correlation coefficient (Pearson correlation, *r*), the values were transferred to standard z scores by Fisher transformation. The connectivity maps of all trial participants were analysed. 

Based on the Automated Anatomical Labelling (AAL) template, pre-processed rs–fMRI data were segmented into 90 regions [29]. A total of 28 out of 90 regions were associated with the functional olfactory cortical networks (OCN) [30]. A mask of the primary and secondary OCN processing areas was created (Appendix A), and the FC spatial maps were presented within the mask.

### 3.7. Interval Assessments and Follow–Up Evaluations

Trial outcomes were measured clinically during follow-up assessment at week 4. Reassessment structural and rs–fMRI brain scans were performed at week 2 (interim) and week 4 (end-of-treatment) for secondary outcome analyses. 

## 4. Outcomes

The primary efficacy outcome was quantitative difference in the measured olfactory function between baseline and end-of-treatment (week 4). Clinical olfactory improvement was defined as a two-point increase in the trial participant’s BTT scores [11].

Secondary outcomes included measured differences in SIT scores, and interval assessments of structural and rs–fMRI neuroradiological changes. Volumetric measurements of bilateral OB/tracts, and MR spectroscopy measurements at the GR and SFC were compared between baseline and end-of-treatment. Seed-based FC analyses within the OCN were compared between groups at three assessment periods: baseline (pre-treatment), interim (week 2), and end-of-treatment (week 4).

## 5. Statistical Analysis

Post hoc sample size calculation was based on the primary efficacy outcome. The minimal sample size required to demonstrate a mean two-point difference in BTT measurements between control and combination groups (with an allocation ratio of 0.5, more than 80% power at a two-sided 5% significance level) was 4 (control group) and 8 (combination group), respectively [11].

The primary efficacy outcomes and differences between means were analysed using Brown–Forsythe and Welch’s analysis of variance (ANOVA) tests. Intergroup evaluations of baseline and end-of-treatment SIT scores were compared using paired *t* test. Categorical variables were compared using Fisher’s exact test.

Two sample *t*-tests were applied to measure the group differences in rs–fMRI with Gaussian random field (GRF) correction (voxel *p* < 0.010, cluster *p* < 0.050, one-tail). Group comparisons of MR spectroscopy data and volumetric measurements of bilateral OB/tracts were performed by Brown–Forsythe ANOVA with Tukey post hoc tests. Correlational analyses were performed using Pearson’s correlation coefficient. Statistical analyses were performed using Prism 9 (GraphPad Software, Boston, MA, USA) and SPSS, version 27 (Chicago, IL, USA). 

## 6. Trial Registration

This study is registered with ClinicalTrials.gov, NCT04900415.

## 7. Data Availability

The data that support the findings of this study are available at the discretion of the corresponding authors, upon reasonable request.

## 8. Results

### 8.1. Clinical and Olfactory Function Status

Between 14 August 2020 and 11 June 2021, 56 individuals were consecutively assessed for eligibility. A total of 26 participants met the eligibility criteria and were randomly assigned to study intervention groups: combination group (*n* = 10), standard care group (*n* = 11), and control group (*n* = 5; Figure 1). Two patients in the standard care group were excluded from analysis due to withdrawal of consent. Two additional patients (one from the combination group, and one from the standard care group) defaulted follow-up after completing the end-of-treatment rs–fMRI brain scans. The neuroradiological results from these two patients were included in the assessments for secondary outcomes, but their clinical data were excluded from the primary analysis. Consequently, 22 trial participants were included in the primary analysis: combination group (*n* = 9), standard care group (*n* = 8), and control group (*n* = 5).

No statistically significant differences were found in the baseline demographics and COVID-19-related OD characteristics between groups (Table 1). The median age of trial participants was 44 years (interquartile range (IQR), 32–57 years), 64% (14 out of 22) were female, and 36% (8 out of 22) had co-existing conditions. Median duration of OD at enrolment was 157 days (IQR 127–175 days). Mean BTT score at baseline and study randomisation was 2.3 (standard deviation (SD), 1.1). There were no differences in smoking status, COVID-19 disease severity, admission SARS-CoV-2 RT-PCR cycle threshold (C_T_) values, duration of OD, and olfactory function test scores measured by BTT and SIT between groups (Table 1).

At end-of-treatment, primary efficacy outcome analysis showed a statistically significant difference in mean BTT scores between groups (*p* < 0.001, Figure 2A). Mean BTT scores were significantly higher for the combination group when compared against the control (*p* < 0.001, difference in means (MD) = 4.4 with 95% confidence interval (CI) 1.7 to 7.2) and standard care groups (*p* = 0.009, MD = 3.2, 95% CI 0.5 to 5.9). There were no differences in BTT scores between standard care and control groups (*p* = 0.229, MD = 1.3, 95% CI −0.9 to 3.4).

In the intragroup comparisons, between baseline and end-of-treatment BTT scores, mean differences of BTT scores were significantly higher for the combination group when compared with the control (*p* = 0.002, MD = 3.3 with 95% CI 1.0 to 5.6) and standard care groups (*p* = 0.012, MD = 2.3, 95% CI 0.3 to 4.2). There was no difference in the mean difference of BTT scores between baseline and end-of-treatment for the standard care and control groups (*p* = 0.199, MD = 1.1, 95% CI −0.9 to 3.0).

In the secondary outcome analysis, there was a statistically significant difference in mean SIT scores between groups (*p* = 0.043) at end-of-treatment. In intragroup comparison, SIT scores were significantly higher in the combination group after treatment (*p* = 0.009, Figure 2B) but no differences were found in the standard care or control groups. Categorisations of SIT results between groups at baseline and end-of-treatment were shown (Figure 2C).

### 8.2. Neuroradiological Outcomes

#### 8.2.1. Seed–Based rs–fMRI Analyses in the Olfactory Functional Network 

In total, 24 trial participants completed all three brain scans: baseline, interim (week 2), and end-of-treatment (week 4). Two participants in the combination group were excluded at baseline due to excess head motions. Therefore, baseline rs–fMRI brain scan analyses were performed for 22 trial participants: combination group (*n* = 8), standard care group (*n* = 9), and control group (*n* = 5). No between-group differences were found in the OCN at baseline. 

At interim analysis (during treatment, week 2), rs–fMRI brain scans revealed significantly higher FC in the combination group than standard care and control groups (Table 2). Significantly higher FC was identified in the right GR (cluster size: 21, peak z value 6.4) in the combination group when compared to the standard care group (Figure 3A).

Combination group showed significantly higher FC (Table 2) in the bilateral anterior cingulate cortex (ACC; left side (cluster size: 38, peak z value 5.0); right side (cluster size: 7, peak z value 4.2)) and left superior temporal gyrus (STG; cluster size: 28, peak z value 5.2) than the control group (Figure 3B). Standard care group also showed higher FC in the left ACC (cluster size: 24, peak z value 4.8; but insignificant difference in the right ACC) and left STG (cluster size: 20, peak z value 6.0) than the control group (Figure 3C). Significantly higher FC in the OCN suggests that combination treatment, with the addition of oral VitA, demonstrated enhanced therapeutic effects in the treatment of persistent OD in long COVID patients.

At end-of-treatment (week 4), combination group showed significantly higher FC in the left inferior frontal gyrus (IFG; cluster size: 21, peak z value 4.2) than the control group (Table 3, Figure 4B). No statistically significant differences were found in other between-group comparisons. In addition, among trial participants in the combination group, positive correlations were found in the increased FC in the left IFG and measured clinical changes in the BTT (*r* = 0.858, *p* < 0.001; Figure 5) and SIT (*r* = 0.548, *p* = 0.042; Appendix A) scores.

#### 8.2.2. MR Spectroscopy Analyses at the Gyrus Rectus and Superior Frontal Cortex

Twenty-four trial participants (combination group (*n* = 10), standard care group (*n* = 9), and control group (*n* = 5)) were included in the MR spectroscopy analyses. At baseline, no statistically significant differences were found in between-group comparisons (*p* = 0.345). 

In the interim scans (week 2), MR spectroscopy showed a statistically significant difference in the NAA/Cr ratio between groups (*p* = 0.021; Appendix A). Importantly, combination group had a significantly higher NAA/Cr ratio (*p* = 0.045) than control group with multiple comparison corrections. At end-of-treatment, significant differences were again demonstrated between groups (*p* = 0.013; Appendix A). Interventional groups showed significantly higher NAA/Cr ratios (combination group, *p* = 0.012; standard care group, *p* = 0.036) than control group. Moreover, positive correlations were also demonstrated between NAA/Cr ratios and olfactory function tests at end-of-treatment (SIT: *r* = 0.644, *p* = 0.001; and BTT: *r* = 0.492, *p* = 0.020; Appendix A) in the combination group.

#### 8.2.3. Volumetric Analysis of the Olfactory Bulbs and Tracts 

No between-group differences were found in the volumetric measurements of the OB/tracts (Appendix A).

## 9. Safety and Tolerability

In all randomly assigned participants, no adverse events were reported. All trial participants completed the assigned intervention without adverse outcomes. 

## 10. Discussion

This is the first, integrated multimodal-multidisciplinary pilot trial evaluating the therapeutic effects of short-course oral VitA in combination with aerosolised diffuser OT (combination treatment) in the management of persistent OD in long COVID patients, when compared with standard care (OT alone) and control (clinical observation). 

The therapeutic effects of smell recovery in combination treatment were confirmed by objective measurable improvements in olfactory function tests (BTT score: *p* < 0.001, MD = 4.4 with 95% CI 1.7 to 7.2); enhanced neural activities in the olfactory functional network (left IFG; cluster size: 21, peak z value 4.2; two-sample *t* test, GRF correction, voxel-level *p* < 0.010, cluster-level *p* < 0.050); and elevated NAA/Cr ratio (*p* = 0.012), as a surrogate marker for viable neurons within the olfactory network (GR and SFC). Positive correlations were found between multiple clinical and neuroradiological parameters in the primary and secondary outcome analyses.

Neurotropic properties of SARS-CoV-2 have been intensely studied [31,32,33]. However, the exact pathogenesis of COVID-19-related OD remains elusive. Human studies demonstrated a high level of astrogliosis (GFAP^+^) and microgliosis (HLA–DR^+^) at the OB during post-mortem examination, but pathologies localised at the ONE, sparing the OB, were also reported (Appendix A) [34,35,36,37]. Furthermore, isolated SARS-CoV-2 infection at the ONE was demonstrated in various animal models without OB involvement (Appendix A) [33,38,39,40,41]. Overall, the currently available evidence suggests that the pathological process of acute COVID-19-related OD may be localised at the ONE [42]. However, the precise pathology of persistent OD in long COVID patients requires further elucidation.

In the treatment against COVID-19-related OD, we postulated that VitA could enhance cellular regeneration at the ONE by promoting stem cell differentiation [43]. Mouse models demonstrated that multipotent horizontal basal cell (HBC) activation and differentiation is governed by the downregulation of Notch1 signalling and ΔNp63α expression. In vitro, p63 expression could be suppressed by retinoic acid (RA; a VitA derivative) in HBC cell culture models, leading to differentiation for CK18^+^ and TuJ1^+^ (neuronal marker) cells, with the latter expressing bipolar morphology, resembling protruding dendrites of olfactory sensory neurons. Furthermore, in vivo stem cell transplantation of RA–treated HBC achieved ONE engraftment, lending further support for VitA treatment in the enhancement of neuroregeneration at the ONE [9,44]. Our findings bolster the potential for VitA as an adjunct in promoting neuronal recovery, thereby expanding its applicability to various olfactory neurosensory disorders and potentially extending its reach to other neuroscience domains, stem cell research, and regenerative medicine beyond the olfactory system.

In this study, end-of-treatment MR spectroscopy revealed elevated NAA/Cr ratio (*p* = 0.012) at the GR and SFC in the combination group when compared to the control. Moreover, higher NAA/Cr ratios at the olfactory apparatus were positively correlated with improved clinical olfactory function tests. As a recognised biomarker for neuronal integrity, elevated NAA/Cr ratios after combination treatment may be indicative for active neuroregeneration. 

In the interim (week 2) rs–fMRI brain scans, bilateral ACC (especially the left ACC) and left STG showed significantly enhanced FC in the combination group. The ACC is a centrally located structure with heterogenous functions and diverse cortical, limbic, and paralimbic connections [45]. While the left STG is a recognised functional structure for auditory, language, and social processing [46,47]. We postulate that superior left-lateralised hemispheric FC enhancements in the ACC and STG during combination treatment may be the result of improved olfactory function, which would have been integrated into the day-to-day neurocognitive functioning of treated patients. 

At end-of-treatment, trial participants in the combination group showed significant improvements in SIT scores (*p* = 0.009), which reflected clinical recovery in odour identification. Furthermore, end-of-treatment rs–fMRI brain scans showed enhanced FC in the left IFG, which was positively correlated with the end-of-treatment SIT scores (*n* = 14, *r* = 0.548, *p* = 0.042). Overall, holistic recovery of smell and FC enhancement in the left IFG after combination treatment are consistent with the known neurocognitive function of the left IFG, where high-level olfactory processing has been associated with episodic memory retrieval, syntactic processing, and semantic processing during olfaction [48,49,50,51].

This pilot study has several limitations. Firstly, the small sample size limits the generalisability of results. The major challenge in recruitment may be related to the reduced prevalence of smell disturbances associated with emerging SARS-CoV-2 variants (30–50% for Alpha and Delta, and approximately 16% for Omicron) [52,53]. Although participants in the combination group were generally younger and had fewer co-morbidities, no statistically significant differences were observed. To confirm the therapeutic efficacy of combination treatment, large-scale, placebo-controlled, double-blind randomised trials are necessary. Nonetheless, this therapeutic approach and management strategy have consistently produced favourable outcomes, from the case report to this pilot study [11].

Secondly, the open-label design might over- or underestimate perceptions of smell change by trial participants. However, the utilisation of multimodal quantitative olfactory function measures (e.g., BTT and SIT) and serial multiplexed neuroradiological evaluations should minimise potential biases. Notably, reporting neuroradiologists were blinded to group allocations and clinical data, ensuring a masked, independent, and consistent assessment of neuroradiological outcomes across all subjects. Moreover, the preliminary results from this study offer valuable effect size estimates for future therapeutic trials, focusing on the quantitative evaluation of olfactory function and rs–fMRI assessments of functional neural networks.

Lastly, the follow–up duration was limited to four weeks. Further analyses with extended evaluation periods are warranted to validate the long-term durability of smell recovery following combination treatment. 

In conclusion, this trial addressed an important unmet need for a large population of long COVID patients suffering from persistent OD, for whom no standardised treatment has been established. Combination treatment, using short-course oral vitamin A and aerosolised diffuser OT, was effective in the treatment for OD. This pragmatic, safe, and affordable treatment strategy could be deployed in an organised national campaign for long COVID patients suffering from persistent neurosensory defects and smell loss.

## Figures and Tables

**Figure 1 brainsci-13-01014-f001:**
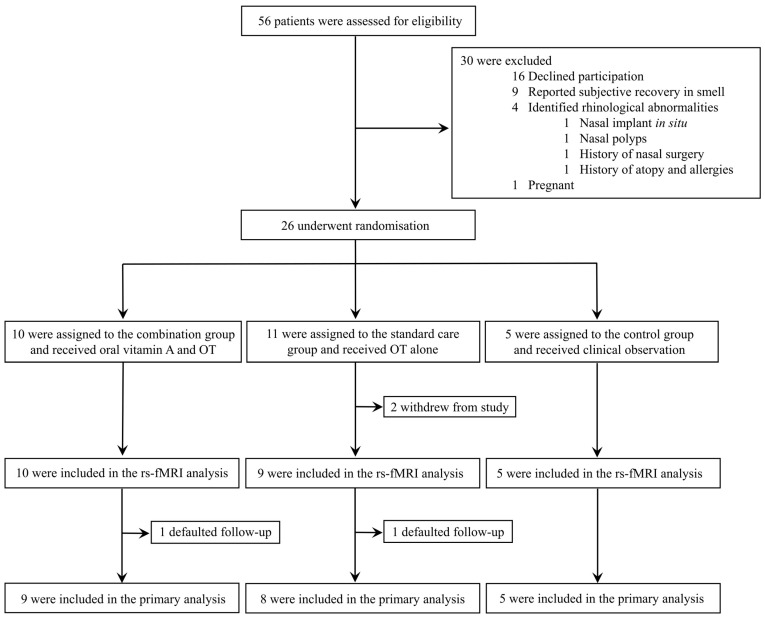
Trial profile. OT: olfactory training; rs–fMRI: resting-state functional magnetic resonance imaging brain scan.

**Figure 2 brainsci-13-01014-f002:**
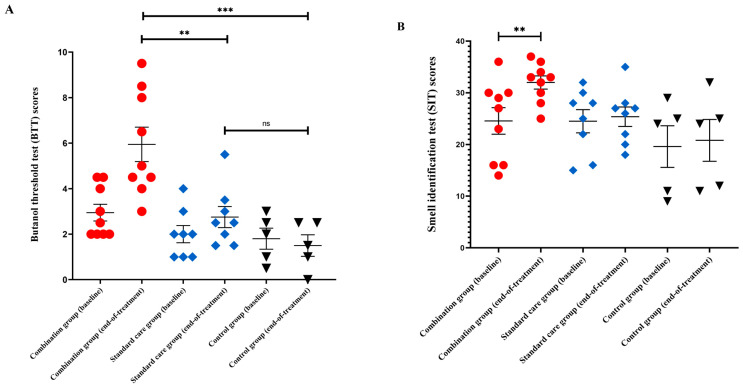
Functional olfactory assessments. (**A**) Scattered plot showing the butanol threshold test (BTT) scores; (**B**) scattered plot showing the smell identification test (SIT) scores; and (**C**) percentage distributions of SIT categories among trial participants, between baseline and end-of-treatment. Error bars are means, and standard error of the mean (SEM). *** *p* < 0.001; ** *p* < 0.010; ns: not significant.

**Figure 3 brainsci-13-01014-f003:**
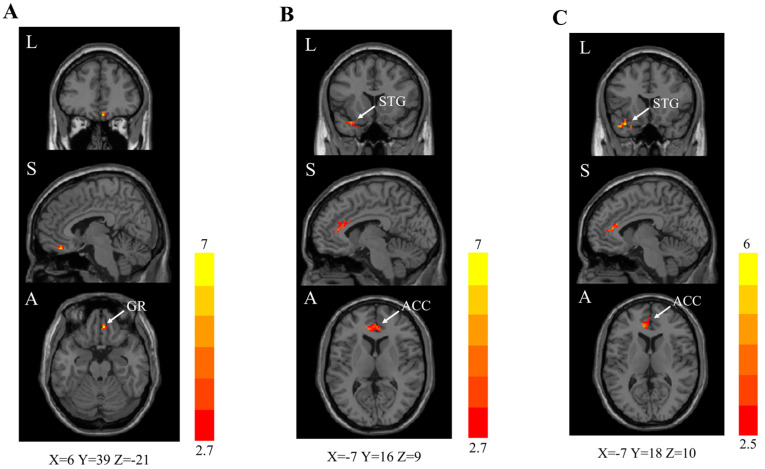
Interim (week 2) rs–fMRI brain scan images (right caudate as the seed region). Two-sample *t* test. GRF correction (voxel-level *p* < 0.010, cluster-level *p* < 0.050). rs–fMRI = resting-state functional magnetic resonance imaging. z values are represented by the colour bars. A = anterior. L = left. S = superior. ACC = anterior cingulate cortex. GR = gyrus rectus. STG = superior temporal gyrus. (**A**) Combination [oral vitamin A in combination with olfactory training (OT)] group versus standard care (OT alone) group. (**B**) Combination group versus control (clinical observation) group. (**C**) Standard care group versus control group.

**Figure 4 brainsci-13-01014-f004:**
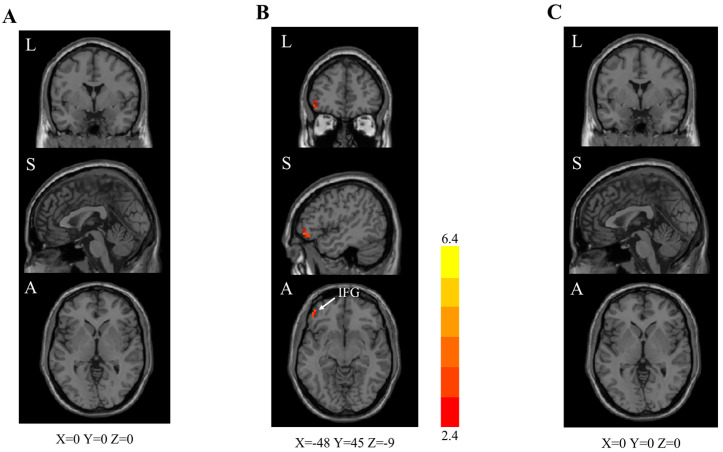
End-of-treatment (week 4) rs–fMRI brain scan images (left caudate as the seed region). Two-sample *t* test. GRF correction (voxel-level *p* < 0.010, cluster-level *p* < 0.050). rs–fMRI = resting-state functional magnetic resonance imaging. z values are represented by the colour bars. A = anterior. L = left. S = superior. IFG = inferior frontal gyrus. (**A**) Combination [oral vitamin A in combination with olfactory training (OT)] group versus standard care (OT alone) group. (**B**) Combination group versus control (clinical observation) group. (**C**) Standard care group versus control group.

**Figure 5 brainsci-13-01014-f005:**
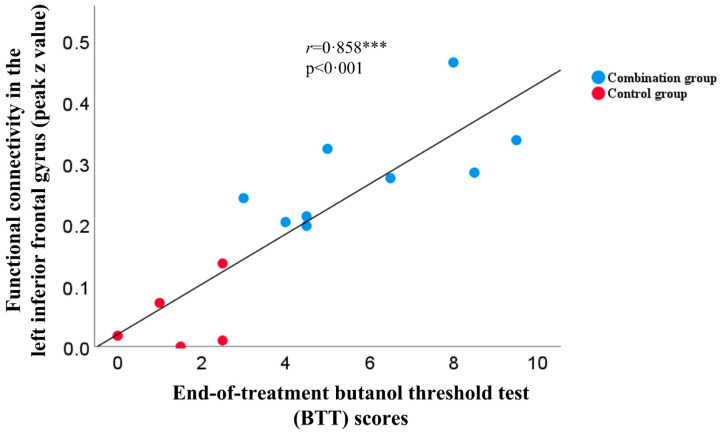
Correlations between functional connectivity in the left inferior frontal gyrus and end-of-treatment BTT scores. The peak z value represents the maximal functional connectivity in the left inferior frontal gyrus (between left caudate seed and voxels) for the combination and control group at the end-of-treatment [two-sample *t* test; GRF correction (voxel-level *p* < 0.010, cluster-level *p* < 0.050)], which demonstrated positive correlation with the end-of-treatment butanol threshold test (BTT) scores (*n* = 14, *r* = 0.858, *p* < 0.001). *** *p* < 0.001.

**Table 1 brainsci-13-01014-t001:** Characteristics of trial participants at baseline.

	Intervention Groups		
Characteristics	Combination (*n* = 9)	Standard Care (*n* = 8)	Control (*n* = 5)
Median age (IQR)–year	36 (26.0–43.0)	49 (37.3–56.3)	58 (50.0–61.0)
Female sex–no. (%)	5 (56)	6 (75)	3 (60)
Tobacco smoker–no. (%)	3 (33)	0 (0)	0 (0)
Coexisting conditions–no. (%)			
Good past health	8 (89)	4 (50)	2 (40)
Diabetes mellitus	0 (0)	0 (0)	1 (20)
Hypertension	0 (0)	1 (13)	2 (40)
Hyperlipidaemia	1 (11)	0 (0)	1 (20)
Atherosclerotic diseases	0 (0)	2 (25)	1 (20)
Rheumatological conditions	1 ^†^ (11)	0 (0)	0 (0)
Malignancies	0 (0)	0 (0)	2 ^‡^ (40)
History of haematological or solid organ transplantation	0 (0)	0 (0)	0 (0)
SARS-CoV-2 diagnosis			
Positive RT–PCR–no. (%)	9 (100)	8 (100)	5 (100)
C_T_ values (IQR)	22.1 (18.1–29.0)	20.1 (15.1–21.7)	17.2 (15.7–22.8)
COVID–19 disease severity			
Mild disease–no. (%)	9 (100)	8 (100)	5 (100)
COVID–19-related OD, onset and duration (IQR)–days			
Onset of OD from first COVID–19 symptom	0.0 (0.0–7.0)	4.0 (1.8–16.0)	3.0 (0.0–3.0)
Median duration of COVID–19-related OD	159 (130.0–163.0)	164.5 (118.3–180.3)	138.0 (135.0–225.0)
COVID–19-related OD, symptomatology–no. (%)			
OD onset			
Sudden	6 (67)	6 (75)	1 (20)
Gradual	3 (33)	2 (25)	4 (80)
OD characteristics			
Anosmia	4 (44)	5 (63)	2 (40)
Hyposmia	4 (44)	2 (25)	3 (60)
Parosmia	1 (11)	3 (38)	0 (0)
Hyperosmia	0 (0)	0 (0)	0 (0)
Phantosmia ^※^	2 (22%)	1 (13)	0 (0)
Olfactory assessments–mean (SD)			
Modified LK score	0.7 (1.0)	0.0 (0.0)	0.0 (0.0)
BTT	2.9 (1.1)	2.0 (1.1)	1.8 (1.0)
SIT	24.6 (7.7)	24.5 (6.3)	19.6 (9.0)
SIT category–no. (%)			
Normosmia	1 (11)	0 (0)	0 (0)
Mild microsmia	1 (11)	2 (25)	0 (0)
Moderate microsmia	3 (33)	2 (25)	1 (20)
Severe microsmia	1 (11)	2 (25)	2 (40)
Anosmia	3 (33)	2 (25)	2 (40)

Combination = oral vitamin A in combination with olfactory training (OT). Standard care = OT alone. Control = clinical observation. ^†^ Rheumatoid arthritis, quiescent disease. ^‡^ Malignancy of the breasts, in remission. ^※^ Phantosmia, detection of rotten substances or burning wood. BTT = butanol threshold test; COVID–19 = coronavirus disease 2019; C_T_ = cycle threshold; Modified LK score = modified Lund-Kennedy endoscopic score; OD = olfactory dysfunction; RT–PCR = reverse transcription–polymerase chain reaction; SARS-CoV-2 = severe acute respiratory syndrome coronavirus 2; SD = standard deviation; SIT = smell identification test.

**Table 2 brainsci-13-01014-t002:** Intergroup analysis of interim (week 2) rs–fMRI brain scans (right caudate nucleus as the seed region).

Olfactory Cortical Network Regions	Cluster Size	Peak z Value
**Combination group vs. standard care group**		
Right gyrus rectus	21	6.4
**Combination group vs. control group**		
Left anterior cingulate cortex	38	5.0
Right anterior cingulate cortex	7	4.2
Left superior temporal gyrus	28	5.2
**Standard care group vs. control group**		
Left anterior cingulate cortex	24	4.8
Left superior temporal gyrus	20	6.0

Two-sample *t* test. GRF correction (voxel-level *p* < 0.010, cluster-level *p* < 0.050). rs–fMRI = resting-state functional magnetic resonance imaging. Combination = oral vitamin A in combination with aerosolised diffuser olfactory training (OT). Control= clinical observation. Standard care = OT alone.

**Table 3 brainsci-13-01014-t003:** Intergroup analysis of rs–fMRI brain scans (left caudate nucleus as the seed region).

Interim Assessment (Week 2)			End-of-Treatment Assessment (Week 4)		
OCN Regions	Cluster Size	Peak z Value	OCN Regions	Cluster Size	Peak z Value
**Combination group vs. control group**					
Right medial frontal gyrus	23	5.6	Left inferior frontal gyrus	21	4.2
**Standard care group vs. control group**					
Right superior temporal gyrus	22	6.1	-	-	-

Two-sample *t* test. GRF correction (voxel-level *p* < 0.010, cluster-level *p* < 0.050). rs–fMRI = resting-state functional magnetic resonance imaging. OCN = olfactory cortical network. Combination = oral vitamin A in combination with aerosolised diffuser olfactory training (OT). Control= clinical observation. Standard care = OT alone.

## Data Availability

The data that support the findings of this study are available at the discretion of the corresponding authors, upon reasonable request.

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
