# Peer review of "A Pilot Study of Short-Course Oral Vitamin A and Aerosolised Diffuser Olfactory Training for the Treatment of Smell Loss in Long COVID"

_brainsci, 2023, doi:10.3390/brainsci13071014_

Round 1

Reviewer 1 Report

This is well conducted study on olfactory dysfunction in COVID. The authors used short–course oral vitamin A and aerosolised diffuser olfactory training for smell loss. The paper in general is well organized and supported with relevant literature. The study and the results are original and novel. Article contents are generally sufficient. The authors presented their results via a well-prepared tables and figures. They discussed it well. It was written as a contribution to the literature.

Reviewer 2 Report

Dear Authors

the Manuscript has a good level of novelty, nevertheless major revisions are necessary.

Major revisions

Abstract: please expand the Background with the aim of the manuscript and reduce the Methods.

Introduction: A limited search on medical publications database permit to obtain several manuscripts on Vitamin A and COVID or on Vitamin A and olfactory training.

Methods: 24 participants were a small number for this study. In fact, Authors wrote “combination treatment (n=10), standard care (n=9), or control (n=5)”; moreover, in table 1 Authors reported 9, 8, 5 participants, respectively, at the end of the selection. The number of participants need to be more discussed because it is limited and not clear.

Methods: Authors did not explain the gender, the median- and the range-age of the participants of each treatment group. In fact, men and female, just like adults with different age, have differences in olfactory dysfunctions development or in their response to COVID treatment. These parameters are not discussed in the Discussion section.

Methods: Authors need to describe the smell identification test SIT used in their study. From Sensonics International there are different SIT: UPSIT, BSIT, etc..

Minor revisions

Abstract: Authors need to avoid (P<0·001, MD=4·4, 95% CI 1·7 to 7·2) and similar numerical data in the Abstract.

Page 1 and 2. In the Abstract and in the Introduction change the sentence “Olfactory dysfunction (OD) is a common neurosensory defect in long COVID patients” because the olfactory dysfunction is not a defect but is a consequence of COVID-19.

Page 1. Please explain “MRI”

All the manuscript: regarding the numerical data, correct all the values in all the manuscript. e.g. “0·001” had to be written as “0.001”.

All the Manuscript: check regarding the words “COVID”, “Covid”, “COVID-19” or “Covid-19” and write it always in the same way.

Introduction: please explain “MR”.

Participants: Authors should indicate the range of age with standard deviation and the number of male and female for each group of treatment and control.

Therapeutic interventions: Change “as described previously” in “as previously described”.

Methods: Authors use the smell identification test SIT, but there are some other tests, like Sniffin’ stick test, used in olfactory studies on COVID. Authors need to write some sentences about it in the Introduction.

Table 1: insert “patients” after each number. The legend needed a revision.

Figure 2: Authors had to explain in the manuscript the differences between “Anosmia, Severe microsmia, etc.. obtained with SIT scores. Moreover, Authors wrote anosmia among the Keywords. In the manuscript there are not sentences that describe anosmia, severe microsmia, etc..

Methods: Author need to insert in Methods paragraph a sentence similar to “data are expressed as mean value +/- standard deviation (SD)”.

Results: Author wrote “Between August 14, 2020, …..”. These sentences are not Results and need to be moved in the Methods paragraph.

Results: mean data with SD had to be written in specific manner.

Insert some sentences discussing recent references, like Li et al. doi: 10.18632/aging.103888,  Stephensen et al. doi: 10.1017/S0007114521000246.

Minor editing of English language required

Reviewer 3 Report

The manuscript (ID: brainsci-2430896) entitled “Short–course oral vitamin A and aerosolised diffuser olfactory training for smell loss in long COVID: a pilot study”, aims to assess clinical improvements of olfactory function in combination treatment when compared to olfactory training alone (standard care) or control (clinical observation).

 The Manuscript evaluated an interesting topic but required major revisions. The Manuscript, even though it is a pilot study, has a low number of patients and controls enrolled, and this represents the main weakness of this randomized–controlled study.

Specific comments:

Introduction

Authors should consider not only quantitative deficits but also qualitative disturbances of smell and/or taste that were found in 35.3% of patients with COVID-19 as reported by Ercoli et al. (Neurological Sciences https://doi.org/10.1007/s10072-021-05611-6).

As regards the reference 13 Authors should indicate that no benefits were found for the use of Vitamin A in the treatment of postinfectious or posttraumatic olfactory loss.

However, other studies (Khani et al., 2021, doi: 10.1016/j.ejphar.2021.174582; Hummel et al., 2017, doi: 10.1007/s00405-017-4576-x) indicated that intranasal vitamin A is beneficial in post-infectious olfactory loss. Consequently, Authors should consider this controversial data in their Introduction.

Material and Methods

Authors should indicate that patients enrolled in this study showed Long COVID-19 with the information of the day range after the negative nasopharyngeal swab.

In addition, mean age and standard deviation for patients and controls should be indicated in the participant’s section. The period of the data collection should be indicated in the Material and Methods section.

My major concern is the low number of subjects enrolled for each sub-group, the combination (n=9), the standard care (n=8), and the control (n=5). The number of patients enrolled in this study is very low and should be implemented because it may create a bias in statistical analyses. Authors indicated this now number as a limitation of this study since it may be due to low incidence of Covid–19 in Hong Kong. Nevertheless, the Centre for Health Protection has recorded in Hong Kong around 1203020-1660455 cases tested positive for Covid–19. Consequently, Authors cannot point to this factor as a limitation of the study and should revise the discussion section.

Minor editing of English language is required

Round 2

Reviewer 2 Report

Dear Authors,

despite the interest in the topic, regarding the Manuscript “Short–course oral vitamin A and aerosolised diffuser olfactory training for smell loss in long COVID: a pilot study”, the low number of participants affect the study so additional experiments needed.

Best regards

Reviewer 3 Report

Since the major limitation of this study is the low number of subjects enrolled for each sub-group, the combination (n=9), the standard care (n=8), and the control (n=5), but Authors did not increase the number of patients enrolled in each group this Manuscript should be rejected. 

Moderate editing of English language is required
